# Bacteriophages—Dangerous Viruses Acting Incognito or Underestimated Saviors in the Fight against Bacteria?

**DOI:** 10.3390/ijms25042107

**Published:** 2024-02-09

**Authors:** Magdalena Podlacha, Grzegorz Węgrzyn, Alicja Węgrzyn

**Affiliations:** 1Department of Molecular Biology, University of Gdansk, Wita Stwosza 59, 80-308 Gdansk, Poland; magdalena.podlacha@ug.edu.pl (M.P.); grzegorz.wegrzyn@biol.ug.edu.pl (G.W.); 2Phage Therapy Center, University Center for Applied and Interdisciplinary Research, University of Gdansk, Kładki 24, 80-822 Gdansk, Poland

**Keywords:** bacteriophages, mammalian immune and nervous system, side effects, dysbiosis, phageome

## Abstract

The steadily increasing number of drug-resistant bacterial species has prompted the search for alternative treatments, resulting in a growing interest in bacteriophages. Although they are viruses infecting bacterial cells, bacteriophages are an extremely important part of the human microbiota. By interacting with eukaryotic cells, they are able to modulate the functioning of many systems, including the immune and nervous systems, affecting not only the homeostasis of the organism, but potentially also the regulation of pathological processes. Therefore, the aim of this review is to answer the questions of (i) how animal/human immune systems respond to bacteriophages under physiological conditions and under conditions of reduced immunity, especially during bacterial infection; (ii) whether bacteriophages can induce negative changes in brain functioning after crossing the blood–brain barrier, which could result in various disorders or in an increase in the risk of neurodegenerative diseases; and (iii) how bacteriophages can modify gut microbiota. The crucial dilemma is whether administration of bacteriophages is always beneficial or rather if it may involve any risks.

## 1. Introduction

Bacteriophages (or, shortly, phages) are viruses infecting bacterial cells and propagating there, using resources present inside their hosts. As such, they are classified as bacterial parasites or parasitoids (rather than predators, as can be found in many papers due to the erroneous recognition of their propagation mode, which is inconsistent with biological definitions of relationships between organisms), depending on details of their life cycles [1].

Phages can propagate according to three general developmental schemes, including lytic, lysogenic, and chronic modes [1,2,3]. The lytic mode consists of the infection of a bacterial host through adsorption of the bacteriophage virion on the cell envelope and injection of its genome (build of RNA or DNA), followed by expression of phage genes, replication of its genome, and finally production and assembly of progeny virions, which are released from the host cell after its lysis (mediated by phage-encoded enzymes). When host cells are under metabolically poor conditions, some bacteriophages may switch to ‘pseudo lysogeny’, where their development is halted, until the metabolism is re-started [4]. The lysogenic mode includes (after adsorption and genome injection) either the integration of the phage genome into the host chromosome and its passive replication together with the nucleoid (as a prophage) or, less frequently, the persistence and replication of phage DNA in the form of a plasmid. However, when the lysogenic host cell is endangered by environmental conditions causing DNA damage, a prophage is excised, and the phage propagation is switched to the lytic mode [5]. The chronic mode (also called a permanent infection) resembles the lytic mode, but newly formed virions are released without killing the bacterial host cell, or at least not killing it immediately. In fact, the host eventually dies, but due to exhaustion of the energy sources rather than to cell lysis [6].

Current calculations strongly suggest that bacteriophages are the most abundant biological objects on Earth [7,8]. As such, their roles in all natural habitats, especially in controlling the spread of bacteria, are enormous [9]. Moreover, there are numerous examples of the importance and the use of bacteriophages in medicine and biotechnology (see below).

In the natural environment, phages control the abundance of bacteria in virtually all habitats, ensuring ecological equilibrium [10]. These viruses facilitate the lateral transfer of bacterial DNA between strains and species, contributing to the modification of microbial cells’ metabolism. Moreover, phage lytic development contributes to the circulation of matter in ecosystems due to bacterial cell lysis and the reallocation of compounds (from cells to the environment) [11]. Recent studies indicated that bacteriophages are involved in multi-organismal interactions, participating in complex microbial trophic networks [12]. Such properties of viruses specific to bacterial hosts led to their employment by people in wastewater monitoring and controlling [13,14].

Bacteriophages also occur in guts of animals and humans, effectively modulating their microbiomes, thus influencing the physiology of these complex eukaryotic organisms [15,16]. Some phages bear genes encoding toxins that enhance or empower bacterial virulence, thus contributing to pathogenicity of their hosts [17,18]. On the other hand, lytic bacteriophages (which ultimately kill their host cells) are being proposed as antibacterial agents to be used in combating infections by pathogenic bacteria, in a treatment called phage therapy [19,20]. A similar approach can be used to protect food, medical devices, and other materials against colonization by bacteria, and especially the formation of a biofilm [21,22].

The importance of bacteriophages in biotechnology is enormous. They may be dangerous for biotechnological processes, as by infecting and destroying bacterial cultures in bioreactors, phages not only disrupt technological processes, but may also thwart the further cultivation of microbes used for the synthesis of bioproducts [23]. On the other hand, these viruses and elements of their genomes are being used to construct various cloning and expression vectors, invaluable tools in genetic engineering [24]. Furthermore, phage display technology, allowing us to expose virtually any peptides on the surface of phage virions when fused to one of coat proteins [25], has revolutionized the biotechnological search for compounds able to bind to any specific target molecules. This technology is useful in a wide spectrum of applications, such as improving medical diagnostics, searching for novel drugs, developing effective vaccines, inactivating toxins, producing novel materials (including nanomaterials), and others [26,27,28,29,30].

The short overview of biology of bacteriophages and they applications, presented above, indicates the enormous importance of these viruses in both nature and human civilization. In this light, understanding their interactions with eukaryotic organisms, including animals and humans, appears to be indispensable for the further development of phage-based systems and technologies employed in medicine and biotechnology.

As mentioned above, bacteriophages are abundant in animal and human intestines [15,16]. Therefore, these complex organisms are exposed to permanent contact with these viruses. Moreover, due to the growing crisis of antibiotic therapy, phages are supposed to provide alternatives in the fight against serious bacterial infections, while in the procedures of the phage therapy, they should be applied in large numbers (usually 10^9^ virions or higher) [19,20]. This raises the question of how bacteriophages are recognized by the animal or human immune system [31].

To make the host immune system ready to respond to a viral infection, a series of complex sequential reactions must take place, which require the activation of specialized proteins, and are key to transitioning the organism into a state of readiness to fight against the virus. The first step is to identify the virus genetic material, whether it is DNA or RNA. In this case, pattern recognition receptors (PRRs) play a key role. Several classes of pattern recognition receptors can be differentiated, including NOD-like receptors, RIG-I-like receptors, C-type lectin receptors, and Toll-like receptors (TLR). The following are responsible for recognizing the genetic material of viruses: TLR9 recognizing DNA, TLR7 and TLR8 identifying ssRNA, and TLR3 recognizing dsRNA [32]. This is followed by an increased production of pro-inflammatory cytokines, especially type I and type III interferons, which induce interferon-stimulated genes (ISG) that exhibit potent antiviral activity [33]. An important component of the immune system that has the greatest (and dual) involvement in the interaction with bacteriophages is the spleen and liver mononuclear phagocyte system. On the one hand, these organs are characterized by the highest accumulation of phage particles, as confirmed by comparative analyses of phage titers from different mammalian internal organs. Nevertheless, the spleen and liver are responsible for the rapid neutralization of bacteriophages, even when a specific immune system response has not been developed [32].

Another crucial part of the overall defense mechanism in the fight against viruses are inflammasomes. Although interleukin-1β (IL-1β) and interleukin 18 (IL-18) are pro-inflammatory cytokines, their initially synthetized pro-forms are biologically inactive and require proteolytic processing with the involvement of caspase-1 which, in turn, requires activation by a multiprotein complex called the inflammasome [33]. Inflammasomes are thus important protein complexes, not only responsible for the maturation of pro-inflammatory cytokines (IL-1β and IL-18), but also for directing cells infected with viral or bacterial pathogens into the path of pyroptosis, the so-called inflammatory mode of cell death. On the other hand, this can help eliminate infections, but in the absence of proper regulation, it can also result in the development of severe inflammation [31].

The cause of the renewed interest of researchers in alternative methods for combating bacterial infections is the problem of antibiotic resistance. However, it cannot be overlooked that, also in the case of therapies based on the applications of bacteriophages, their efficacy in repeated administration or long-term use may be impaired by the complex mechanisms of bacterial resistance that are constantly evolving, representing a kind of arms race that poses a serious challenge to modern medicine and veterinary medicine. One strategy used by bacteria is to modify the surface molecules that phages use as specific receptors for adsorption. These modifications include, but are not limited to, mutations that lead to disruption of the synthesis of these molecules, resulting in their absence or a fundamental change in their structure. However, it is a double-edged weapon of sorts, not only protecting bacteria from phages, but also capable of causing them to be significantly weakened. This is because these molecules often determine important processes, such as motility, cell membrane integrity, or nutrient transportation [34]. Importantly, the use of therapies based on two or more bacteriophages appears to be an effective solution in the fight against bacterial resistance to phage therapy. There are two key arguments in favor of this approach. First, there is a low probability of several mutations causing resistance to different bacteriophages in the same cell. Second, the balance of benefits and losses could be unfavorable for the bacterial cell in the event of acquiring resistance to several phages, due to disruption or loss of key functions resulting from the occurrence of the above-mentioned mutations [35]. The situation becomes more complicated when a phage cocktail consists of bacteriophages using the same target receptors on the bacterial cell surface, in which case the phenomenon of cross-resistance can occur, giving the bacteria the advantage of being resistant to several phages simultaneously. It is, therefore, also important to consider the range and potential diversity of receptors on host cells that phages present in the cocktail recognize, and their interactions during adsorption. The administration strategy is also important. For example, the risk of developing bacterial resistance can be reduced by replacing the simultaneous administration of all phages with sequential dosing or being preceded by the use of an appropriate antibiotic [36]. However, the mechanisms of development of bacterial resistance to different combinations of therapeutic agents and frequency of therapeutics used are still under intensive investigation [37].

A specific issue is the problem of bacterial infections of the gastrointestinal tract. Although phage therapy may be effective in the eradication of the infectious bacteria, the question arises as to whether significant changes in the composition of the gut microbiota can be recognized as an adverse effect that can induce long-term negative consequences for the organism. Phage therapy still raises a lot of ambiguities and concerns, especially because of the need to look for alternatives in fighting infections with antibiotic-resistant strains of bacteria. It is crucial to assess whether bacteriophages are our allies, or Trojan horses that can stimulate the immune system in an uncontrolled way, disrupt the functioning of the nervous system, and modify the composition of the intestinal microbiota.

Therefore, the aim of this review is to answer the question of how animal/human immune systems respond to bacteriophages under physiological conditions and under conditions of reduced immunity. We will also discuss the issue of whether bacteriophages can induce negative changes in brain functioning after crossing the blood–brain barrier (BBB), which could result in various disorders or increase the risk of neurodegenerative diseases. Moreover, we will describe and discuss how phages can modify gut microbiota, and whether it is always beneficial, or rather if it may involve serious risks.

## 2. General Safety and Efficacy of Phage Therapy

Considering the use of phage therapy, one should take into account the impact of bacterial toxins which, as a result of the lysis of bacterial cells by bacteriophages, affect human/animal immune systems. To address this problem, a review of the available literature has been conducted, including both case reports (35 cases) and results from animal experiments (20 reports). They suggested that phage therapy is generally safe and well-tolerated. However, due to the lack of sufficient data, the negative effects of bacteriophage interactions with eukaryotic cells cannot be unequivocally ruled out, especially as there are references indicating that phages can stimulate the human immune system in ways that are not fully understood so far [38]. It was also demonstrated that, although bacteriophages modulate the immune response in a significant way, this modulation is not associated with harmful side effects [39]. Such a conclusion was made on the basis of a study that aimed to compare the efficacy and safety of various treatments of the mouse model of acute pneumonia, induced by two *Escherichia coli* strains (536 and LM33). Apart from two bacteriophages (536_P1 and LM33_P1), administered intranasally, three antibiotics (ceftriaxone, cefoxitin and imipenero-cilastatin) were tested. Interestingly, both bacteriophages and antibiotics were effective in controlling the infection, and the rapid lysis of bacterial cells following phage therapy did not induce an enhanced innate inflammatory response. Moreover, blood parameters normalized more promptly as a result of the phage therapy, compared to antibiotics. Only slight increases in lung interferon-γ (IFN-γ) and interleukin-12 (IL-12) levels was induced by bacteriophage 536_P1 administration [39].

In another study, a 20-day experiment, using an analogous model of pneumonia in mice, confirmed that the phage cocktail against *Staphylococcus aureus* NOVO12 (P68 and K710 phages), applied topically, did not induce adverse effects. A safety analysis of phage therapy included mucosal changes, tissue damage, or infiltration of the immune cells [40]. In addition, experiments with a large mammalian model were also conducted to test the safety and effectiveness of phage therapy [41]. After 7 days of biofilm formation, sheep with sinusitis were treated with a phage cocktail CT-PA, which was administered twice a day, by frontal trephine flushes, for one week. A detailed histopathological analysis confirmed the lack of adverse changes in the sinuses, lungs, heart, liver, spleen, and kidneys. Moreover, phage therapy was shown to be highly effective in eradicating the bacterial biofilm.

Many studies on phage therapy focused on its effectiveness, sometimes marginalizing aspects of the safety or potential interactions with mammalian immune systems, which can be negative with long-term use. In this regard, not only the route of administration, but also the location of the infection, which determines the potential direction of bacteriophage propagation to adjacent tissues/organs, can be crucial.

Potential interactions with eukaryotic cells are also dependent on whether monotherapy or a phage cocktail is used [42]. Various animal models were used in different studies, from the most common with mice and rats, to rabbits, dogs, sheep, or pigs. The main problem in interpreting and evaluating the results obtained is that the induced infection is usually acute rather than chronic, which is quite a limitation. The literature data indicated that monotherapy can be effective for skin infections, but the optimal results are obtained with a phage cocktail containing phages with an appropriate host range [43].

It has been demonstrated experimentally how important choosing the appropriate route of administration is, and thus to consider the pharmacokinetics of bacteriophages in assessing their therapeutic potential [44]. Those studies were conducted using mice that were infected with a high dose of *Pseudomonas aeruginosa*, followed by verification of the efficiency of a cocktail consisting of three bacteriophages, which was administered once by three different routes: intramuscular, intraperitoneal, or subcutaneous. As expected, the intraperitoneal administration proved to be the most effective, providing the opportunity for the rapid spreading of the bacteriophages throughout the organism. Another research approach was based on a single administration of bacteriophages, but with an appropriately selected ratio of phage particles to bacterial cells (multiplicity of infection, m.o.i.) [45]. Namely, topically applied bacteriophages were used in a mouse model of skin wounds. It was shown that a single administration of bacteriophages at an m.o.i. of 200 was more effective than all the synthetic preparations tested or the commonly recommended antibiotics. However, the use of phage preparations with a high m.o.i. raises questions about the potential loss of efficiency with repeated use in short intervals (due to production of anti-phage antibodies), and about the unpredictable effects of possible interactions with eukaryotic cells.

As far as the respiratory system is concerned, reports on phage therapy in the control of upper respiratory bacterial infections are very few, in contrast to studies focusing on lower respiratory infections, which may be due to their greater seriousness or sometimes even risk to the patient’s life [42]. In this case, the available literature data indicate that the effectiveness of phage therapy is largely determined by the choice of the most suitable route of administration. Intranasal application seems to be the most appropriate one. However, a major limitation hindering the objective assessment of both the efficacy and safety of phage therapy as an alternative treatment of respiratory infections is the lack of adequate animal models.

When analyzing the safety issues of phage therapy, it is also worth asking the fundamental question of why the immune system does not treat bacteriophages as classic, potentially dangerous viruses. It is estimated that more than 31 billion bacteriophages undergo transcytosis from the intestine to different areas of our organism every day. In addition to being commonly found in the gut, bacteriophages have been confirmed in saliva, urine, and peripheral blood, and have also been shown to propagate to various internal organs [46]. Because bacteriophages and their products are non-self-antigens, the immune system is able to recognize and react in a way that, theoretically, reduces the benefits of bacteriophage therapy. The presence of antibodies neutralizing the bacteriophage T4, hosted by *E. coli*, were found in more than 80% of investigated persons, despite the fact that they had never received phage therapy [47].

Bacteriophages are generally assumed to be non-toxic, but there are reports indicating their immunomodulatory properties, notably their anti-inflammatory potential or their ability to activate phagocytosis. The recruitment of neutrophils, necessary for effective phage therapy against *P. aeruginosa*, is also an interesting example [46]. As mentioned above, the viral genetic material is recognized as PAMPs by TLRs. In the case of phage DNA, it can be postulated that after transcytosis or phagocytosis, it is recognized by TLR9. The result of the interaction of bacteriophages with immune cells is the synthesis of cytokines, which is particularly important during bacterial infections. However, it is essential to realize that the anti-inflammatory effect is not always the same as the immunosuppressive potential of the drugs or agents used. In the case of some phages, we may be dealing with the creation of special conditions in which the immune response is weakened, directly translating into the virulence of the bacteria and their viability [32]. Even if cytokine balance is disturbed as a result of phage therapy, it is short-lived and normalizes considerably faster than in the case of antibiotic therapy, the adverse effects of which are long-lasting and threaten the homeostasis of the entire organism. In the case of antibiotics, the most frequently reported adverse effects are those affecting the gastrointestinal tract, such as nausea, vomiting, diarrhea, abdominal pain, loss of appetite, bloating, and gut flora disturbances. These effects also include, depending on the dose used and frequency of the intake, skin allergies, skeletal pains, and dysfunctions of the reproductive and nervous systems. Antibiotics also affect the composition of the intestinal microbiota which, in turn, influences the immune response, especially in children which, in the future, translates into susceptibility to infections and the development of allergies and autoimmune diseases. Compared to adults, the antibiotic treatment of infants has disproportionately greater consequences because the infant microflora is evolving, quite unstable, and immature until 2–3 years of age [39]. Therefore, it is so important to know all potential risks associated with administered therapeutics, even if their effectiveness has been known for many years, as well as to constantly search for new, equally effective but safer alternative methods.

## 3. The Use of Bacteriophages in Animals/Humans with Efficient Immune Systems during Bacterial Infection

It is widely believed that bacteriophages are unable to infect eukaryotic cells. This is due to a number of limitations. First of all, the structural elements of the phage tail only allow it to bind to specific receptors on the surface of the target bacterial cell (despite the possibility of an unspecific binding to some molecules present on the surface of mammalian cells). Moreover, there are differences between the eukaryotic and prokaryotic molecular machineries that determine the proper course of the transcription, translation, and replication processes. Nevertheless, the fact that bacteriophages are able to exert a direct effect on the mammalian immune system cannot be denied. The effect is anti-inflammatory, and includes the impact on innate immunity via the modulation of the cytokine response, as well as adaptive immunity through antibody production [48]. This was corroborated by the results of recent experiments, which have shown that a phage cocktail administered orally for 14 days to *Salmonella enterica*-infected chickens exhibited an anti-inflammatory potential, evident in the form of increased concentrations of cytokines showing anti-inflammatory effects, especially interleukin-10 and interleukin-4 (IL-10 and IL-4). The efficacy of the phage cocktail involved not only effective elimination of the bacteria, but also a reduction of otherwise strong inflammation, measured as elevated levels of key pro-inflammatory cytokines (IL-1β, IL-6, IFN-γ, IL-8, and IL-12), which normalized when the phages were administered 24 h after the bacterial infection or 2 days after detection of the tested bacteria in feces. Moreover, administration of the phage cocktail did not disrupt the number of lymphocytes and their subpopulations, which are key cells of the immune system, not only in the fight against bacterial or viral infections, but also in the organism’s daily functioning. The results not only confirmed the effectiveness of phage therapy, but also provided an important rationale for confirming the safety of its use in terms of the immune system, as well as in relation to the functioning of the stress axis, which determines the homeostasis of the entire organism [49].

The analyses were not limited to white blood cell parameters. It was shown that bacterial infection also has negative consequences for hematological indicators, which can quickly lead to deterioration of the overall health of the animals. However, sufficiently rapid administration of a phage cocktail can result in the effective and safe normalization of hematocrit, MCV, MCH, and MCHC. When analyzing the safety of administering various drugs or potential therapeutics, the hepatotoxic effects must also be taken into account. Unlike antibiotic therapy, phages turned out to not cause any adverse effects, estimated on the basis of the lack of increased levels of some enzymes, like alanine transaminase and aspartate aminotransferase, which indicated proper liver functioning [50].

Gangwar et al. [51] confirmed the lack of adverse effects of phage therapy after both single and repeated (28 days) oral administration of the *Klebsiella pneumoniae*-specific XDR bacteriophage to rats. Animals receiving a daily dose of 10^10^ plaque-forming units (p.f.u.)/mL or 10^15^ p.f.u./mL (an extremely high dose) of bacteriophage did not differ from the control group in terms of the analyzed parameters. Verification of the safety of phage therapy included hematological and biochemical parameters, as well as the cytokine profile (IL-1β, IL-4, IL-6, and IFN-γ).

The analyses of the cytokine profiles, pharmacokinetics, and biodistribution after the administration of various doses of three bacteriophages were also conducted in experiments with healthy rats and monkeys [52]. The pharmacokinetic profile was characterized by a significant decrease in the bacteriophage titer in plasma after intravenous administration in both rats and monkeys, but the key determinants were the type of bacteriophage and the dose used. The administered bacteriophages were eliminated from the organism within 72 h. The innate immune response was assessed on the basis of concentrations of cytokines in plasma. Elevated concentrations of TNF-α, IL-6 and keratinocyte chemoattractant/growth-regulated oncogene were observed in rats after the administration of the tested bacteriophages. However, it should be noted that normalization of these values occurred within 24 h, which is still significantly shorter than the long-term disruption of cytokine balance observed after antibiotic therapy. Interestingly, the results of the analysis of the cytokine profile in nonhuman primates were different, which is of key importance in the context of testing the safety of phage therapy intended for use in medicine, and not only in veterinary medicine. Namely, there were no increases in the concentrations of TNF-α, IL-6, IL-10, and IFN-γ after the administration of the SE_SZW1 phage. The same bacteriophage caused increases in the concentrations of the tested cytokines in the plasma of rats. Therefore, it is crucial to note that when analyzing the safety of phage therapy, many variables must be taken into account, including the type of bacteriophage, the route of administration, the site of application, the frequency of administration, and the species of the animal study model.

Other important factors may also be the differences between the immune responses in males and females. These might arise from the influence of sex hormones, as well as other factors, like differences in the permeability of the BBB. Studies conducted with Salmonella-specific phages have shown a lack of adverse effects after two weeks of daily oral administration of the cocktail composed of two bacteriophages (vB_SenM-2 and vB_Sen-TO17) to mice of both sexes. The safety was demonstrated not only in the context of immune functions, but also the functions of the central nervous system, indirectly tested through behavioral pattern analyses. Interestingly, the dysfunctions of both the aforementioned systems were observed in females after antibiotic therapy [53].

As the organism’s response to viral infections is hormonally modulated, including a gender factor in this type of analysis may be crucial, indeed. The hormonal cycle affects at least the induction of PRRs which, in turn, significantly modulates the production of IFN-β or MX2, which are proteins that have a significant impact on the course of viral infections [54].

As is evident from the above-mentioned examples, the question of assessing the safety of phage therapy in normally functioning animal (or human) organisms, especially during bacterial infection, is not easy. The problem is even more complicated when the organism is weakened by a severe and/or aggravating chronic disease.

## 4. The Use of Bacteriophages in Animals/Humans with Impaired Immune Systems during Infection

It appears that, in the context of the development of phage therapy, major therapeutic approaches are focused on the treatment of chronic infections that affect the skin and lungs. In fact, these conditions are especially difficult to treat, due to impairment of the immune system caused by various reasons, like long-term exposure to attacks of antibiotic-resistant, pathogenic bacteria in the respiratory system or tissue damage arising after burns of the skin [55].

Acute bacterial lung infections are often caused by *P. aeruginosa*. Although bacteriophage Pf, specific to this bacterium, was used as an excellent model in studies demonstrating how phages are able to modify the evolutionarily highly conserved mechanisms of the mammalian immune response to various types of infections [55], its use in phage therapy is unlikely. Namely, Pf is a temperate phage, and it can even promote bacterial pathogenesis.

*P. aeruginosa*, in addition to causing severe skin infections or lung diseases, hinders the healing of wounds resulting from burns. In this light, it is worth mentioning the first clinical trial with phage therapy, where a phage cocktail was applied topically to 13 patients hospitalized due to skin burns. It turned out that, in this case, phage therapy was generally of poor efficacy, evidently less effective than the standard treatment (1% sulfadiazine silver emulsion cream). Additionally, 3 of the 13 patients receiving the phage cocktail reported adverse effects [56]. However, it is important to note that a very low dose of bacteriophages (100 p.f.u., which is several orders of magnitude lower than usually used doses) was administered. Thus, it is clear that such low doses are not efficient in the phage therapy, and it is crucial to prepare phage cocktails containing sufficient titers of bacteriophages, like 10^9^ p.f.u./mL or higher.

A more complex situation concerns patients with severe bacterial infections and a number of comorbidities. This can be exemplified by the case of a 60-year-old patient, hospitalized due to peritonitis caused by *Enterobacter cloacae*, as well as abdominal sepsis, dispersed intravascular coagulation, herniation, and bowel strangulation. As a result of prolonged hospitalization, this patient developed difficult-to-heal bedsores, colonized by *P. aeruginosa*, as well as kidney damage. Despite the introduction of phage therapy by the intravenous route, which successfully improved renal parameters, the patient died [57].

Social and economic life has recently come to a standstill as a result of the COVID-19 pandemic, which has left almost 7 million people dead worldwide. The major reason for such a high mortality rate was the disruption of the communication between innate and adaptive immunity, resulting in the cytokine storm, accompanied by an inability to produce antibodies to neutralize the virus in a timely manner. However, another reason appeared to be the secondary bacterial respiratory infections that occurred in many patients. It was suggested that the special weapon in the fight against both problems may be bacteriophages [58]. As confirmed by the results of in vitro and in vivo studies, carried out over recent years, phages can influence the course infections caused by eukaryotic viruses. These effects of phages on viral infections in animals/humans involve not only the bacteriophages themselves, but also other compounds present in phage lysates (contaminations if the lysates are not highly purified) or phage genetic material. These antiviral properties include the synthesis of interferons by animal/human cells in the presence of bacteriophages, the affinity of bacteriophages for the same cellular receptors as those specific to eukaryotic viruses, and the induction of the release of antibodies that cross-react with pathogenic viruses [59]. The possibility for the use of modified bacteriophages, through the “phage display” technique, to produce specific antibodies against SARS-CoV-2, should not be overlooked, giving patients much-needed time to produce their own specific antibodies and protecting the organism from the fatal consequences of a heightened immune response, especially the huge overproduction of cytokines [58].

As already mentioned, secondary respiratory infections, accompanying viral diseases, are a serious problem. Analyses of samples of blood and those taken from the respiratory tract of COVID-19 patients that suffered from secondary infections showed that, in most cases, the pathogenic agents belonged to the ESKAPE group, i.e., *Enterococcus faecium*, *S. aureus*, *Klebsiella pneumoniae*, *Acinetobacter baumannii*, *P. aeruginosa*, and *Enterobacter* spp. Unfortunately, in many cases, last-resort antibiotics are administered, such as colistin, which cause many adverse effects, like damage of organs and tissues, long-term changes in the intestinal microbiome, and/or increased antibiotic resistance [60]. Therefore, the search for alternative methods should be a priority. Phage therapy with bacteriophages specific to these bacteria is a promising option.

It must be noted that the effective use of bacteriophages in combating bacterial infections strongly depends on the availability of phages capable of destroying specific strains of pathogenic microorganisms. Therefore, the specific pathogen should be isolated first and tested for its susceptibility to phages present in the collection of the hospital or clinical center. This implies that any effective, systemic phage therapy procedures must be based on the presence of large collections of bacteriophages that are specific to various strains of many pathogenic bacteria. The commonly occurring phenomenon of the high specificity of bacteriophages (which is otherwise beneficial for not causing significant changes in the microbiome while eliminating pathogens), in this aspect, might by identified as one of limitations of phage therapy. Namely, one can easily imagine a situation where the available phage collection (if not large enough) lacks specific strain(s) capable of infecting the bacterial isolate identified as an etiologic agent of the disease. One possible solution is the close cooperation of different centers possessing phage collections, enabling the quick exchange and sharing of bacteriophage strains of requested specificity.

## 5. Bacteriophages and the Central Nervous System

For many years, the brain was considered as an immunologically privileged organ. This was primarily due to the lack of the immune system surveillance, and thus, the fact that, in healthy individuals, there are no lymphocytes (which are immunologically competent cells), natural killer (NK) cells (characterized by cytotoxic activity, especially against tumor cells), and antibodies in the brain. The situation is completely different in the case of injuries, infections, or autoimmune diseases. The result of these events is partial damage, or sometimes complete destruction, of the BBB, allowing the free flow of immune cells. Even when the integrity of the BBB is preserved, antigen-stimulated lymphocytes can penetrate from the peripheral blood into the brain tissue, mainly through adhesion molecules, integrins, and selectins. The inflammation accompanying these processes is characterized by an increased inflow of pro-inflammatory cytokines, such as TNF-α and IFN-γ. One should also consider the reactive microglia which, under the influence of cytokines, can successfully act as antigen-presenting cells [61]. Therefore, in a healthy organism, the brain is characterized by apparent immune privileging, due to the absence of lymphatic vessels and the presence of the BBB.

Importantly, bacteriophages are also present in the central nervous system, as a result of their ability to penetrate the BBB [62]. This raises the crucial question of whether this is a beneficial phenomenon, or a serious cause for concern.

Recent analyses of the efficacy and safety of phage therapy in the rat model also included assessments of the functions of the brain [63]. The analyses were performed primarily to test cognitive processes, locomotor activity, and the functioning of the hippocampus, a brain structure crucial for memory. It was demonstrated that the intraperitoneal administration of bacteriophages (10^8^ p.f.u./mL, administered for seven consecutive days) did not show any adverse effects on memory processes, the functioning of hippocampal neurons (CA1, CA2 and CA3 hippocampus areas), or the parameters tested in blood serum.

Similarly, it was demonstrated that phage therapy did not negatively affect the functioning of the central nervous system, which was reflected by the lack of disturbances in the analyzed behavioral patterns in mice of both sexes. However, after antibiotic therapy, many adverse effects were noted, both in relation to key parameters of the immune system and those of the nervous system, which were particularly severe in females [53].

The relationships between the gut phageome, the gut bacteriome, and executive functions (which also include cognitive processes such as initiation, planning, action sequencing, working memory or cognitive flexibility) have been verified in light of the use of phage therapy [64]. The complex experimental scheme included studies with oral administration of bacteriophages to mice and fruit flies (*Drosophila melanogaster*). Both direct administration of bacteriophages to fruit flies and transplantation of the gut microbiota in mice resulted in improved memory processes through impacts on the expression of genes crucial for synaptic plasticity, neuronal development, or memory formation. These results suggested that, by modifying the diet and the gut microbiota, a highly beneficial effect on cognitive processes can be achieved.

Neurodegenerative diseases have very complex mechanisms, often not yet fully understood. A good example is Parkinson’s disease with a multifactorial background. Among the factors responsible for its development, many researchers point to the involvement of pathogens that enter the brain from the respiratory tract, or those that are the part of the intestinal microbiota. Some of them were thought to induce severe inflammation, which would then lead to neurodegeneration of the brain [65]. Some reports indicated that the gut microbiome of Parkinson’s disease patients has unusually high number of dairy-borne bacteriophages. It was speculated that they could cause elevated concentrations of central pro-inflammatory cytokines and also auto-antibodies, which are particularly toxic to dopaminergic neurons [66]. It has been shown that this group of bacteriophages (especially phage 936) could intensify both peripheral and central inflammation, and negatively influence behavioral patterns, which was particularly evident in older individuals and deepened with time [67]. On the other hand, the proposal of the enhancement of the neurodegeneration by bacteriophages is in contrast to the results described in the preceding paragraph, indicating benefits, rather than dysfunctions, in the brain after the use of phage therapy.

It is worth emphasizing that many important biomarkers identified in the brain are also present in the intestines, which seems to support the hypothesis regarding the role of the gut–brain axis in the pathogenesis of neurodegenerative disorders. Moreover, inflammation results in intestinal dysbiosis, which has serious pathological consequences, often leading to intensified inflammatory processes in the periphery which, if left untreated, even propagate to the central nervous system, resulting in neuroinflammation [67]. However, it remains to be determined what the role of bacteriophages in these processes is, and whether the phage-mediated effects are positive or negative for brain functions.

An important issue is also that if phages are used as therapeutics agents in organs where they have to reach, like the brain (which could be achievable after crossing BBB by bacteriophages transported by blood), their stability must be taken into consideration. Indeed, different phages vary in their blood lifetime. When four different phages were tested in mice, their titers in blood at 2–4 h after administration varied by several orders of magnitude [68]. On the other hand, one should note that bacteriophages were suggested to be able to contribute to microbe-triggering of amyloid-associated neurodegenerative diseases. More detailed studies suggested that this mechanism has a potential origin not in phages themselves, but in phage DNA packaging-generated capsids that are not mature [69,70].

## 6. Bacteriophages and Gut Microbiota Modulation

Over the last few years, knowledge about the role of intestinal microbiota in maintaining homeostasis and the development of many disorders resulting from increased intestinal permeability and, as a consequence, inflammation, has been systematically increasing. There is a direct relationship between dysbiosis, inflammation, and the development of various pathological conditions, such as neurodegenerative diseases, cardiovascular disorders, cancer, diabetes, rheumatoid arthritis, or progressive aging processes. Due to the fact that there are approximately 10^15^ bacteriophages in the human intestine, which is 10 times more than the number of bacterial cells and 100 times more than the number of all human cells in the organism, we cannot ignore the fact that bacteriophages are one of the main determinants of the composition of the intestinal microbiota. When phage cocktails containing bacteriophages specific to several species of bacteria were added to rats’ drinking water, changes in the microbiome and impaired gut permeability were observed. Therefore, it was speculated that phages can be treated as a specific category of mammalian pathogens [71]. Again, this was quite a controversial proposal, especially in the light of studies demonstrating the benefits of applications of bacteriophages.

In this light, an extensive study was conducted, in which the effects of a cocktail of bacteriophages infecting *E. coli* on the intestinal microbiota, and markers of intestinal and peripheral inflammation in healthy people, were investigated [72]. The phage cocktail was administered orally for 28 days, and analyses of stool and blood samples included markers of inflammation, lipid metabolism, and gut microbiota composition. Apart from the elimination of *E. coli*, there were no major changes in the composition of the microbiome, as only minor changes in some bacterial species were noted. These included a slight increase in the abundance of *Eubacterium* spp. and a decrease in the proportion of taxa closely related to *Clostridium perfringens*. However, there was no effect of the tested phage cocktail on other parameters, except for a small but statistically significant decrease in IL-4 concentrations in peripheral blood. It can, therefore, be concluded that if the therapy is based on phage cocktails with a narrow host spectrum, the aim of which is to eliminate specific strains of bacteria, the risk of serious disturbances in the composition of the intestinal microbiota will be limited.

Although it is estimated that the global ratio of bacteriophage virions to bacterial cells on Earth is 10 to 1, in the human gut, this ratio may be considerably lower, indicating the different nature of the interactions and strategies employed by both phages and bacteria. A growing body of evidence suggests that phages are crucial to maintaining the mammalian organism’s homeostasis and health, as their number and diversity are considerably modified in the course of various diseases, like inflammatory bowel disease and type 1 diabetes [73]. Indeed, the composition of the phageome can change very dynamically, which correlates significantly with the organism’s health status.

Changes in the composition of the microbiome are readily apparent, as could be demonstrated in patients suffering from ulcerative colitis, where mucosa-inhabiting bacteriophages infecting *E. coli* and other *Enterobacteria* were predominant, while the general abundance of caudate phages decreased [74]. Similarly, the composition of the gut microbiome in patients with metabolic syndrome was analyzed [75]. In the latter case, the dominance of *Streptococcaceae*- and *Bacteroidaceae*-infecting phages, and a significant reduction in the abundance of *Bifidobacteriaceae*-infecting bacteriophages, were shown.

A significant difference in the gut microbiome of Crohn’s disease patients relative to control subjects was also noted. The stool samples from the patients showed a considerable predominance of bacteriophages infecting bacteria from *Lactococcus*, *Enterococcus*, and *Lactobacillus* genera [76].

The general conclusion is that the composition of the gut microbiome, including the phageome, changes over the course of different diseases. Thus, it was postulated that its analysis can provide a useful biomarker for indicating the risk of a particular disorder, monitoring its progression, or verifying the effectiveness of potential therapies [77]. However, it is still not clear whether the changes in the composition and abundance of bacterial species, as well as these parameters of bacteriophages occurring in the gut, are the causes of the effects of various diseases. Therefore, in our opinion, without understanding the roles of microbes in specific diseases and the mechanisms of their actions in different disorders, the use of the microbiome content as a disease marker would be premature. Only when the specific mechanisms leading to particular disorders are recognized, and roles of bacteriophages and microbial cells in these mechanisms are understood, can one propose and develop reliable disease markers.

## 7. Concluding Remarks

Despite a great deal of research into their biology, mechanisms of action, and potential interactions with eukaryotic cells, bacteriophages still seem to hold many secrets, the understanding of which will allow their enormous potential to be realized. They may prove to be an effective weapon not only to combat antibiotic-resistant bacteria, but also in the fight against various other diseases, like viral infections. On the other hand, the roles of bacteriophages in neurodegenerative diseases, cancers, or metabolic disorders are still unclear, and even the determination if their effects are beneficial or destructive remains to be performed. Definitely, in terms of their safe use in medicine and veterinary medicine, we need to fully understand biology of bacteriophages in order to effectively minimize the risk of potential adverse effects, as well as to take advantage of all the possibilities of the use of phage therapy. The major fields of interactions between bacteriophages and mammalian organisms in the light of phage therapy are summarized in Figure 1.

## Figures and Tables

**Figure 1 ijms-25-02107-f001:**
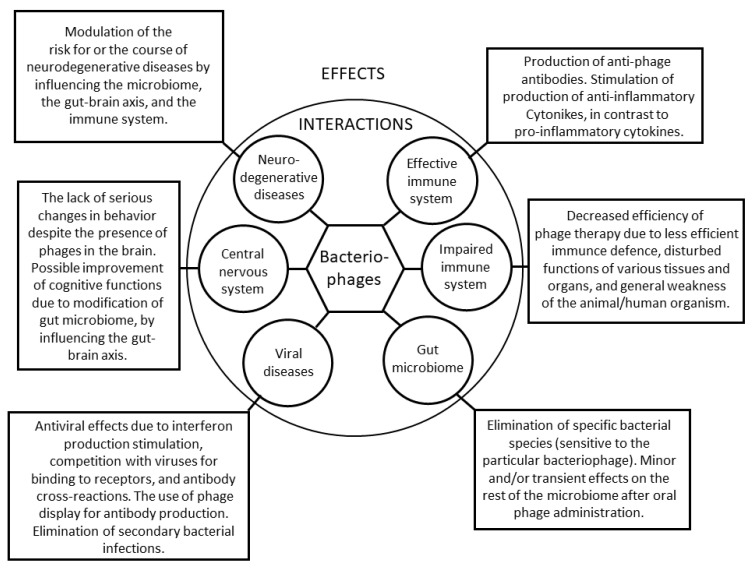
Schematic representation of the major interactions of bacteriophages with animal/human organisms, with special emphasis on administration of these viruses during phage therapy. The scheme is presented as a Cyske rosette chart (according to Cyske et al. [78]), where the interactions between bacteriophages and specific systems (like the immune system, gut microbiome, or central nervous system) or stages (like viral infections or neurodegenerative diseases) are indicated in the central part of the figure, while their effects are described in the peripheral squares.

## Data Availability

Not applicable.

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
