# Peer review of "Bacteriophages—Dangerous Viruses Acting Incognito or Underestimated Saviors in the Fight against Bacteria?"

_ijms, 2024, doi:10.3390/ijms25042107_

Round 1
Reviewer 1 Report
Comments and Suggestions for Authors
The authors have made an attempt to give a review on the topic “Bacteriophages – dangerous viruses acting incognito or underestimated saviors in the fight against bacteria?”. However in order to help readers better understand the review, the authors can provide a brief explanation on phage life cycle, phage diversity, their role in nature – animal health, human health and maintenance of ecosystem. This can make the article more explicit to any scientist not specifically working on phage. The introduction can thus be modified accordingly.
For the moment, the manuscript requires a revision to effectively communicate the information in a review or summarized form. The manuscript also needs a through revision to check for grammatical errors and typographical errors.
1. Italicize the scientific names of the microorganisms throughout the manuscript
2. In last paragraph of the introduction the authors begin the sentence as “the aim of the review is to answer the question”…. When it is already mention that it is a question the paragraph need not use question mark for the same. It can be written as statements.
3. The authors under the sub-head – General safety and efficacy of the phage therapy have listed finding from numerous publication which are of high relevance. However, a comprehensive summary drawn from all the data mention would add value to the review.
4. The authors mention it is widely believed that the bacteriophages are unable to infect eukaryotic cells. Give reference to the statement. What about mycophage and phycophages?
Comments on the Quality of English Languagethe manuscript requires a revision to effectively communicate the information in a review or summarized form. The manuscript also needs a through revision to check for grammatical errors and typographical errors
Author Response
REVIEWER’S COMMENT:
The authors have made an attempt to give a review on the topic “Bacteriophages – dangerous viruses acting incognito or underestimated saviors in the fight against bacteria?”. However in order to help readers better understand the review, the authors can provide a brief explanation on phage life cycle, phage diversity, their role in nature – animal health, human health and maintenance of ecosystem. This can make the article more explicit to any scientist not specifically working on phage. The introduction can thus be modified accordingly.
RESPONSE:
As requested by the reviewer, an introduction to bacteriophage biology (life cycles, diversity, roles in nature and maintenance of ecosystem, roles in animal and human health) and applications in biotechnology has been included. The new text in present in the revised manuscript, lines 27-89.
REVIEWER’S COMMENT:
For the moment, the manuscript requires a revision to effectively communicate the information in a review or summarized form. The manuscript also needs a through revision to check for grammatical errors and typographical errors.
RESPONSE:
The text has been corrected as requested.
REVIEWER’S COMMENT:
Italicize the scientific names of the microorganisms throughout the manuscript
RESPONSE:
This has been corrected as indicated by the reviewer.
REVIEWER’S COMMENT:
In last paragraph of the introduction the authors begin the sentence as “the aim of the review is to answer the question”…. When it is already mention that it is a question the paragraph need not use question mark for the same. It can be written as statements.
RESPONSE:
This has been corrected as indicated by the reviewer.
REVIEWER’S COMMENT:
The authors under the sub-head – General safety and efficacy of the phage therapy have listed finding from numerous publication which are of high relevance. However, a comprehensive summary drawn from all the data mention would add value to the review.
RESPONSE:
As requested by the reviewer, a summary of this chapter has been included. The new text in present in the revised manuscript, lines 245-283.
REVIEWER’S COMMENT:
The authors mention it is widely believed that the bacteriophages are unable to infect eukaryotic cells. Give reference to the statement. What about mycophage and phycophages?
RESPONSE:
By definition, bacteriophages are viruses infecting bacterial and archaeal (prokaryotic) cells. Obviously, mycophages and phycophages are viruses of eukaryotic (micro)organisms, however, they are not classified as bacteriophages sensu stricto. Therefore, the statement present in the manuscript, and mentioned by the reviewer, is still valid.
Reviewer 2 Report
Comments and Suggestions for Authors
In their review Podlacha et al. describe the effects of bacteriophage therapies on the homeostasis and dysbiosis of bacteria as well as the safety when using such therapies. This is an intereting and timely review and should be published. A few points should be considered before acceptance. In detail:
1. Line 36/37: as we do not know how phages enter phagocytes and are processed, this is a severe oversimplification and should be reformulated.
2. In general: it is important to know how the phages are administered. This is not always clear from most of the text. It should be added every time an experiment is described.
3. Line 178/179: 1010 or 1015 phages are claimed to be applied. I am sure the authors meant 1010 and 1015. Please correct also ad other locations.
4. What always concerns me is the acquisition of resistance of bacteria to phages. This apparently is avoided by applying cocktails of phages. The authors might want to explain this in the text.
Author Response
In their review Podlacha et al. describe the effects of bacteriophage therapies on the homeostasis and dysbiosis of bacteria as well as the safety when using such therapies. This is an interesting and timely review and should be published. A few points should be considered before acceptance. In detail:
REVIEWER’S COMMENT:
- Line 36/37: as we do not know how phages enter phagocytes and are processed, this is a severe oversimplification and should be reformulated.
RESPONSE:
As recommended by the reviewer, this problem has been discussed in more detail. The new text in present in the revised manuscript, lines 101-114.
REVIEWER’S COMMENT:
- In general: it is important to know how the phages are administered. This is not always clear from most of the text. It should be added every time an experiment is described.
RESPONSE:
The text has been completed according to the reviewer’s recommendation.
REVIEWER’S COMMENT:
- Line 178/179: 1010 or 1015 phages are claimed to be applied. I am sure the authors meant 1010 and 1015 Please correct also ad other locations.
RESPONSE:
The text has been completed according to the reviewer’s recommendation.
REVIEWER’S COMMENT:
- What always concerns me is the acquisition of resistance of bacteria to phages. This apparently is avoided by applying cocktails of phages. The authors might want to explain this in the text.
RESPONSE:
As recommended by the reviewer, this issue has been discussed in more detail. The new text in present in the revised manuscript, lines 125-155.
Reviewer 3 Report
Comments and Suggestions for Authors
This is a brilliant review of an important field of under-developed promise, written by a strong microbiology group at the University of Gdansk. Although phage therapy of antibiotic-resistant bacterial infections have been proposed and studied for decades, the untoward host response of the phage particles has barely received attention. Thus, the authors’ choice of this issue is highly appropriate and timely. The Abstract is by far the most appealing part of the review, as it clearly describes the full rationale of this review and sets the tone for the rest of the article.
I have several suggestions / comments that, if implemented, should improve the review further.
1) In Sections, each topic is logically titled and comprehensively covers the pros and cons of all aspects of phage therapy. My general concern is that the main body of each topic is essentially a summary description of all available papers, with little or no mechanistic insights. Perhaps there were no mechanistic query in those papers, but a good review should attempt to provide some, even by intellectual speculation. The authors actually did that early on, when mentioning the PAMPs such as phage DNA. This could be continued in all the sub-Sections.
2) A similar comment goes for antibiotics. When writing about the “General safety and efficacy of phage therapy”, specially in line 84-89, it would be nice of any negative effects (so-called ‘side effects’) of antibiotics are known that could be added (plus their mechanisms). This would further strengthen the case for phage therapy. Just one paragraph, summarizing the main points would be enough. Specifically, are there studies that indicated an untoward effect of antibiotics on host immunity, cytokine response, etc? Allergy to penicillin, for example, is common.
Minor English errors: These are only a few examples, but there may be others.
Line 49: On the other hand, despite this can help eliminate infection…
Lines 63, 66: Two “Moreover”s so close to each other do not sound nice. Rephrase one.
Line 80: Such a conclusions was made…
Overall excellent English, with only a few typographical errors, which the authors are fully capable of fixing. I have pointed to a few in my review.
Author Response
This is a brilliant review of an important field of under-developed promise, written by a strong microbiology group at the University of Gdansk. Although phage therapy of antibiotic-resistant bacterial infections have been proposed and studied for decades, the untoward host response of the phage particles has barely received attention. Thus, the authors’ choice of this issue is highly appropriate and timely. The Abstract is by far the most appealing part of the review, as it clearly describes the full rationale of this review and sets the tone for the rest of the article.
I have several suggestions / comments that, if implemented, should improve the review further.
REVIEWER’S COMMENT:
- In Sections, each topic is logically titled and comprehensively covers the pros and cons of all aspects of phage therapy. My general concern is that the main body of each topic is essentially a summary description of all available papers, with little or no mechanistic insights. Perhaps there were no mechanistic query in those papers, but a good review should attempt to provide some, even by intellectual speculation. The authors actually did that early on, when mentioning the PAMPs such as phage DNA. This could be continued in all the sub-Sections.
RESPONSE:
As recommended by the reviewer, each section is now finished with a discussion and summary, including some suggestions and speculations on the mechanisms of discussed issues. These concluding paragraphs are present in following fragments of the manuscript: lines 256-283, 363-366, 431-444, 504-512, and 564-575.
REVIEWER’S COMMENT:
- A similar comment goes for antibiotics. When writing about the “General safety and efficacy of phage therapy”, specially in line 84-89, it would be nice of any negative effects (so-called ‘side effects’) of antibiotics are known that could be added (plus their mechanisms). This would further strengthen the case for phage therapy. Just one paragraph, summarizing the main points would be enough. Specifically, are there studies that indicated an untoward effect of antibiotics on host immunity, cytokine response, etc? Allergy to penicillin, for example, is common.
RESPONSE:
According to the reviewer’s recommendation, adverse effects of antibiotics are described in the revised manuscript (lines 270-283).
REVIEWER’S COMMENT:
Minor English errors: These are only a few examples, but there may be others.
RESPONSE:
The language has been checked and corrected where necessary.
REVIEWER’S COMMENT:
Line 49: On the other hand, despite this can help eliminate infection…
RESPONSE:
This has been corrected as indicated by the reviewer.
REVIEWER’S COMMENT:
Lines 63, 66: Two “Moreover”s so close to each other do not sound nice. Rephrase one.
RESPONSE:
This has been corrected as indicated by the reviewer.
REVIEWER’S COMMENT:
Line 80: Such a conclusions was made…
RESPONSE:
This has been corrected as indicated by the reviewer.